# Efficient Personalized Text-to-image Generation by Leveraging Textual Subspace

## Abstract

Personalized text-to-image generation has attracted unprecedented attention in the recent few years due to its unique capability of generating highly-personalized images via using the input concept dataset and novel textual prompt. However, previous methods solely focus on the performance of the reconstruction task, degrading its ability to combine with different textual prompt. Besides, optimizing in the high-dimensional embedding space usually leads to unnecessary time-consuming training process and slow convergence. To address these issues, we propose an efficient method to explore the target embedding in a textual subspace, drawing inspiration from the self-expressiveness property. Additionally, we propose an efficient selection strategy for determining the basis vectors of the textual subspace. The experimental evaluations demonstrate that the learned embedding can not only faithfully reconstruct input image, but also significantly improves its alignment with novel input textual prompt. Furthermore, we observe that optimizing in the textual subspace leads to an significant improvement of the robustness to the initial word, relaxing the constraint that requires users to input the most relevant initial word. Our method opens the door to more efficient representation learning for personalized text-to-image generation.

## 1 Introduction

An important human ability is to abstract multiple visible concepts and naturally integrate them with known visual content using a powerful imagination (Cohen et al., 2022; Ding et al., 2022; Gao et al., 2021; Kumar et al., 2022; Li et al., 2022; Skantze & Willemsen, 2022; Zhou et al., 2022). Recently, a method for rapid personalized generation using pre-trained text-to-image model has been attracting public attention (Gal et al., 2022; Kumari et al., 2022; Ruiz et al., 2022). It allows users to represent the input image as a "concept" by parameterizing a word embedding or fine-tuning the parameters of the pre-trained model and combining it with other texts. The idea of parameterizing a "concept" not only allows the model to reconstruct the training data faithfully, but also facilitates a large number of applications of personalized generation, such as text-guided synthesis (Rombach et al., 2022b), style transfer (Zhang et al., 2022), object composition (Liu et al., 2022), etc.

As the use of personalized generation becomes more widespread, a number of issues have arisen that need to be addressed. The problems are two-fold: first, previous methods such as Textual Inversion (Gal et al., 2022) choose to optimize directly in high-dimensional embedding space, which leads to inefficient and time-consuming optimization process. Second, previous methods only target the reconstruction of input images, degrading the ability to combine the learned embedding with different textual prompt, which makes it difficult for users to use the input prompt to guide the pre-trained model for controllable generation. A natural idea is to optimize in a subspace with high text similarity[1], so as to ensure that the learned embedding can be flexibly combined with textual prompt. Meanwhile, optimizing in the low-dimensional space can also improve training efficiency and speed up convergence. However, existing methods directly used gradient backpropagation to

---

[1]As stated in (Gal et al., 2022), high text similarity indicates that encoding the embedding into the pre-trained model is more likely to generate the corresponding images (e.g., input the embedding of the word "cat" will output a photo of cat), which is usually measured by the cosine similarity between the image and text features transformed by the CLIP model (Hessel et al., 2021).

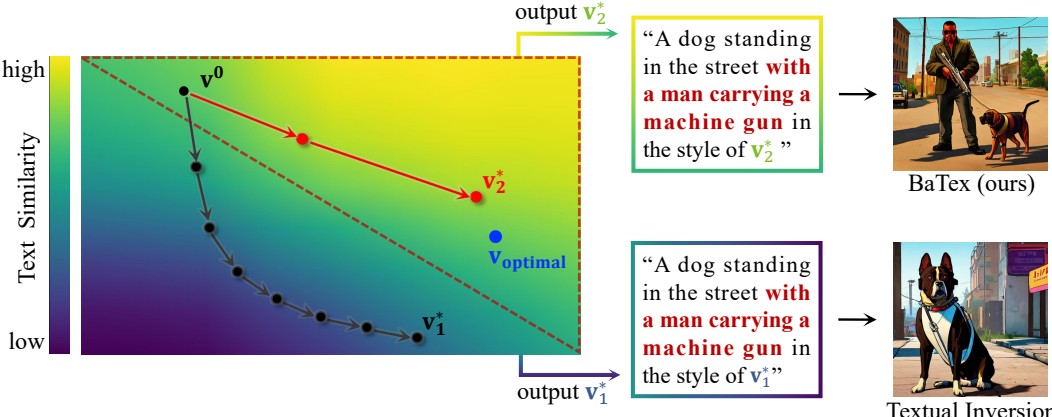

Figure 1: Method Comparison of Textual Inversion and BaTex. Red dashed area denotes the textual subspace $\mathbb{S}$.

optimize the embedding (Gal et al., 2022), making it difficult to explicitly constrain the embedding to a specific subspace for each dataset.

In order to bypass this difficulty, we drew inspiration from the self-expressiveness property[2], which led us to the realization that any target embedding $\mathbf{v}$ can be reconstructed by the combination of other pre-trained embeddings $\mathbf{v}_1, \mathbf{v}_2, \ldots, \mathbf{v}_M$ in the vocabulary $V$ provided by text-to-image models, where $M < |V|$ and $|V|$ is the number of elements in $V$. Once such embeddings $\mathbf{v}_1, \mathbf{v}_2, \ldots, \mathbf{v}_M$ are obtained, we can construct a subspace $\mathbb{S}$ by spanning them, i.e., $\mathbb{S} = \text{span}(\mathbf{v}_1, \mathbf{v}_2, \ldots, \mathbf{v}_M)$. Subsequently, we can conduct an efficient optimization in a low-dimensional space $\mathbb{S}$. Specifically, we explicitly select a number of semantically relevant embeddings from the vocabulary as $\mathbf{v}_1, \mathbf{v}_2, \ldots, \mathbf{v}_M$, and then we form a semantically meaningful subspace $\mathbb{S}$. To achieve semantically relevant embeddings, we introduce a rank-based selection strategy that uses the nearest neighbour algorithm. This strategy efficiently selects $\mathbf{v}_1, \mathbf{v}_2, \ldots, \mathbf{v}_M$ that are semantically close to the input concept, which allows the embeddings learned by our method to be naturally combined with other texts.

To this end, we proposed a method BaTex, which can efficiently learn arbitrary embedding in a textual subspace[3]. Comparing with existing methods like such as Textual Inversion (Gal et al., 2022), the proposed BaTex do not require to search solutions in the entire high-dimensional embedding space, and thus it can improve training efficiency and speed up convergence. The schematic diagram is shown in Figure 1, where BaTex optimizes in a low-dimensional space (shown in red dashed area), which requires fewer training steps and achieves higher text similarity.

We have experimentally demonstrated the expressiveness and efficiency of the proposed BaTex across multiple object and style concepts. BaTex completely avoids the overfitting issues observed in some previous methods (Kumari et al., 2022; Ruiz et al., 2022). In addition, it has the potential to integrate with more advanced and expressive large-scale text-to-image models such as Stable-Diffusion-2[4]. By efficiently exploring the embedding in a textual subspace, our method opens the door to further advancements in efficient personalized text-to-image generation. We summarize the main contributions as follows:

- We propose a novel method BaTex for learning arbitrary embedding in a low-dimensional textual subspace, which is time-efficient and better preserves the text similarity of the learned embedding.

---

[2]Each data point in a union of subspaces can be efficiently reconstructed by a combination of other points in the dataset. (Elhamifar & Vidal, 2013)

[3]Subspace with high text similarity.

[4]https://huggingface.co/stabilityai/stable-diffusion-2

- On the theoretical side, we demonstrate that the selected basis embedding can be combined to produce arbitrary embedding. Also, we derive that the proposed method is equivalent to applying a projection matrix to the update of embedding of Textual Inversion.

- We experimentally demonstrate that the learned embeddings can not only faithfully reconstruct the input image, but also significantly improve its alignment with different textual prompt. In addition, the robustness against initial word has been improved, relaxing the constraint that requires users to input the most relevant word as initialization.

## 2 BACKGROUND AND RELATED WORK

In this section, we give the background and related work about deep generative models and its extensions. We first introduce the diffusion models, a class of deep generative models, then we briefly discuss text-to-image synthesis and personalized generation.

### 2.1 DIFFUSION MODELS

The goal of deep generative models, such as Flow-based Models (Dinh et al., 2016; Du et al., 2022), VAE (Burgess et al., 2018), GAN (Goodfellow et al., 2020) and Diffusion Models (Dhariwal & Nichol, 2021), is to approximate an unknown data distribution by explicitly or implicitly parameterizing a model distribution using the training data.

As a class of deep generative models which has been shown to produce high-quality images, Diffusion Models (Dhariwal & Nichol, 2021; Ho et al., 2020; Nichol & Dhariwal, 2021; Song et al., 2020) synthesize data via an iterative denoising process. As suggested in (Ho et al., 2020), a reweighted variational bound can be utilized as a simplified training objective and the sampling process aims to predict the corresponding noise added in forward process. The denoising objective is finally realized as a mean squared error:

$$\mathcal{L}_{original} = \mathbb{E}_{x_0,c,\epsilon,t}[\|\epsilon - \epsilon_\theta(\alpha_t x_0 + \sigma_t \epsilon, c)\|_2^2], \tag{1}$$

where $x_0$ are training data with conditions $c$, $t \sim \mathbb{U}(0,1)$, $\epsilon \sim \mathbb{N}(0, \mathbf{I})$ is the gaussian noise sampled in the forward process, $\alpha_t, \sigma_t$ are pre-defined scalar functions of time step $t$, $\epsilon_\theta$ is the parameterized reverse process with trainable parameter $\theta$.

Recently, an introduced class of Diffusion Models named Latent Diffusion Models (Rombach et al., 2022a) raises the interest of the community, which leverages a pre-trained autoencoder to map the images from pixel space to a more efficient latent space that significantly accelerates training and reduces the memory. Latent Diffusion Models consist of two core models. Firstly, a pre-trained autoencoder is extracted, which consists of an encoder $\mathcal{E}$ and a decoder $\mathcal{D}$. The encoder $\mathcal{E}$ maps the images $x_0 \sim p(x)$ into a latent code $z_0$ in a low-dimensional latent space. The decoder $\mathcal{D}$ learns to map it back to the pixel space, such that $\mathcal{D}(\mathcal{E}(x)) \approx x$. Secondly, a diffusion model is trained on this latent space. The denoising objective now becomes

$$\mathcal{L}_{latent} = \mathbb{E}_{z_0,c,\epsilon,t}[\|\epsilon - \epsilon_\theta(\alpha_t z_0 + \sigma_t \epsilon, c)\|_2^2], \tag{2}$$

where $z_0 = \mathcal{E}(x_0)$ is the latent code encoded by the pre-trained encoder $\mathcal{E}$.

For conditional synthesis, in order to improve sample quality while reducing diversity, *classifier guidance* (Dhariwal & Nichol, 2021) is proposed to use gradients from a pre-trained model $p(c|z_t)$, where $z_t := \alpha_t z_0 + \sigma_t \epsilon$. *Classifier-free guidance* (Ho & Salimans, 2022) is an alternative approach that avoids this pre-trained model by instead jointly training a single diffusion model on conditional and unconditional objectives via randomly dropping $c$ during training with probability $1 - w$, where $w$ is the guidance scale offering a tradeoff between sample quality and diversity. The modified predictive model is shown as follows: $\hat{\epsilon}_\theta(z_t, c) = w\epsilon_\theta(z_t, c) + (1 - w)\epsilon_\theta(z_t, \phi)$, where $\phi = c_\theta(\text{“ ”})$ is the embedding of a null text and $c_\theta$ is the pre-trained text encoder such as BERT (Devlin et al., 2018) and CLIP (Radford et al., 2021).

### 2.2 TEXT-TO-IMAGE SYNTHESIS

Recent large-scale text-to-image models such as Stable-Diffusion (Rombach et al., 2022a), GLIDE (Nichol et al., 2021) and Imagen (Saharia et al., 2022) have demonstrated unprecedented semantic

generation. We implement our method based on Stable-Diffusion, which is a publicly available 1.4 billion parameters text-to-image diffusion model pre-trained on the LAION-400M dataset (Schuhmann et al., 2021). Here $c$ is the processed text condition.

Typical text encoder models include three steps to process the input text. Firstly, a textual prompt is input by the users and split by a tokenizer to transform each word or sub-word into a token, which is an index in a pre-defined language vocabulary. Secondly, each token is mapped to a unique text embedding vector, which can be retrieved through an index-based lookup (Gal et al., 2022). The embedding vector is then concatenated and transformed by the CLIP text encoder to obtain the text condition $c$.

### 2.3 Personalized Generation

As the demand for personalized generation continues to grow, personalized generation has become a prominent factor in the field of machine learning, such as recommendation systems (Amat et al., 2018) and language models (Cattiau, 2022). Within the vision community, adapting models to a specific object or style is gradually becoming a target of interest. Users often wish to input personalized real images to parameterize a "concept" from them and combine it with large amounts of textual prompt to create new combination of images.

A recent proposed method Textual Inversion (Gal et al., 2022) choose the text embedding space as the location of the "concept". It intervenes in the embedding process and uses a learned embedding $\mathbf{v}$ to represent the concept, in essence "injecting" the concept into the language vocabulary. Specifically, it defines a placeholder string $S$ (such as "A photo of $*$") as the textual prompt, where "$*$" is the pseudo-word corresponding to the target embedding $\mathbf{v}$ it wishes to learn. The embedding matrix $y \in \mathbb{R}^{N \times d}$ can be obtained by concatenating $\mathbf{v}$ with other frozen embeddings (such as the corresponding embeddings of "a", "photo" and "of" in the example), where $N$ is the number of words in the placeholder string[5] and $d$ is the dimension of the embedding space. The above process is defined as (Gal et al., 2022): $y \leftarrow Combine(S, \mathbf{v})$.

The optimization goal is defined as: $\arg\min_{\mathbf{v}} \mathbb{E}_{z_0,\epsilon,t}[\|\epsilon - \epsilon_\theta(\alpha_t z_0 + \sigma_t \epsilon, c_\theta(y))\|_2^2]$, where $z_0$, $\epsilon$ and $t$ are defined in Equation (1) and (2). Please note that $y$ is a function of $\mathbf{v}$. Although Textual Inversion can extract a single concept formed by 3-5 images and reconstruct it faithfully, it can not be combined with textual prompt flexibly since it solely considers the performance of the reconstruction task during optimization. Also, it searches the target embedding in the high-dimensional embedding space, which is time-consuming and difficult to converge.

To address the issues above, we observe that the pre-trained embeddings in the vocabulary is expressive enough to represent any introduced concept. Therefore, we explicitly define a projection matrix to efficiently optimize the target embedding in a low-dimensional tetxual subspace, which speeds up convergence and better preserves the text similarity of the learned embedding.

## 3 The Proposed BaTex Method

In this section, we introduce the proposed BaTex method. Following the definition of Stable Diffusion (Rombach et al., 2022a), $\mathbb{E} = \mathbb{R}^d$ is the word embedding space with dimension $d$ and $V$ is the word vocabulary defined by the CLIP text model (Radford et al., 2021). The words in vocabulary $V$ corresponds to a set of pre-trained vectors $\{\mathbf{v}_i\}_{i=1}^{|V|}$, where $|V|$ is the cardinality of set $V$.

### 3.1 Optimization Problem

We first state that any vector in the embedding space $\mathbb{E}$ can be represented by the embeddings in the vocabulary $V$, as shown in the following theorem whose proof can be found in Appendix B.

**Theorem 1** *Any vector $\mathbf{v}$ in word embedding space $\mathbb{E}$ can be represented by a linear combination of the embeddings in vocabulary $V$.*

---

[5]In practice, the embedding matrix $y \in \mathbb{R}^{N_{max} \times d}$ where $N_{max}$ is the pre-defined maximum number of words in a sentence and other $N_{max} - N$ words are filled with terminator. For clarity of expression, we only consider the words with practical meaning in the main text.

---

**Algorithm 1** Selection Strategy of Textual Subspace.

---

**Input:** vocabulary $V$, initialization embedding $\mathbf{u}$, dimension of the embedding space $d$, dimension of textual subspace $d_1$, vector distance $\mathcal{F}$
**Phase 1: reordering the embeddings using the vector distance**
Calculate the distance vector $dist_V = (\mathcal{F}(\mathbf{u}, \mathbf{v}_1), \ldots, \mathcal{F}(\mathbf{u}, \mathbf{v}_{|V|}))^T$
Order the embeddings $(\mathbf{v}_{i_1}, \ldots, \mathbf{v}_{i_{|V|}}) \leftarrow Order((\mathbf{v}_1, \ldots, \mathbf{v}_{|V|}), dist_V)$
**Phase 2: rank-based selection strategy**
**repeat**
    Select a proper number $M$, get the top $M$ embeddings: $\mathbf{v}_{i_1}, \ldots, \mathbf{v}_{i_M}$
    Form the embedding matrix: $V_M \leftarrow [\mathbf{v}_{i_1}, \ldots, \mathbf{v}_{i_M}]$
    Compute the rank of the embedding matrix: $r(V_M)$
**until** $r(V_M) \geq d_1$
**Output:** embeddings $\mathbf{v}_{i_1}, \ldots, \mathbf{v}_{i_M}$

---

As stated in Theorem 1, any vector in $\mathbb{E}$ can be represented by a linear combination of the embeddings in the vocabulary $V$. Now we define the weight vector $\mathbf{w} = (w_1, \ldots, w_{|V|})^T$ with each component corresponding to a embedding in $V$, and the embedding $\mathbf{v} = \sum_{i=1}^{|V|} w_i \mathbf{v}_i$. Since the users input an initial embedding $\mathbf{u} \in V$, we wish the start point in $\mathbb{E}$ to be the same as $\mathbf{u}$ in our algorithm. Thus, we initialize the weights as:

$$\mathbf{w}^0 := (w_1^0, \ldots, w_{|V|}^0) = (0, \ldots, 0, 1|_{i=i_\mathbf{u}}, 0, \ldots, 0)^T, \tag{3}$$

where $i_\mathbf{u}$ denotes the index corresponding to $\mathbf{u}$. Then the embedding $\mathbf{v}$ to be learned can now be initialized as:

$$\mathbf{v}^0 = w_1^0 \mathbf{v}_1 + \cdots + w_{i_\mathbf{u}}^0 \mathbf{u} + \cdots + w_{|V|}^0 \mathbf{v}_{|V|}. \tag{4}$$

Subsequently, it can be combined with the placeholder string to form the embedding matrix $y$ as stated in Section 2.3. The reconstruction task can be formulated as the following optimization problem:

$$\arg\min_{\mathbf{v} \in \mathbb{E}} \mathcal{L}_{rec} := \mathbb{E}_{z_0, \epsilon, t}[\|\epsilon - \epsilon_\theta(\alpha_t z_0 + \sigma_t \epsilon, c_\theta(y))\|_2^2]. \tag{5}$$

where $z_0$, $\epsilon$, $t$, $\alpha_t$ and $\sigma_t$ are detailed in Equation (1) and (2), $c_\theta$ is the text encoder defined in Section 2.1, $y$ is a function of $\mathbf{v}$. To solve problem (5), we iteratively update the weight vector $\mathbf{w}$ by using gradient descent with initial point $\mathbf{w}^0$, so that the embedding $\mathbf{v}$ is also updated.

While it is expressive enough to represent any concept in the embedding space $\mathbb{E}$, most weights and vectors are unnecessary since the rank $r(A_V) = d$ as stated in Theorem 1, where $A_V = [\mathbf{v}_1, \ldots, \mathbf{v}_{|V|}] \in \mathbb{R}^{d \times |V|}$. Thus, we only need at most $d$ vectors to optimize with $\mathbf{u}$ included, which corresponds to selecting $d$ linearly-independent vectors $\mathbf{v}_{i_1}, \mathbf{v}_{i_2}, \ldots, \mathbf{u}, \ldots, \mathbf{v}_{i_d}$ from vocabulary $V$.

### 3.2 TEXTUAL SUBSPACE

As detailed in Section 3.1, any vector in the embedding space $\mathbb{E}$ can be obtained using $d$ vectors $\mathbf{v}_{i_1}, \mathbf{v}_{i_2}, \ldots, \mathbf{u}, \ldots, \mathbf{v}_{i_d}$. However, optimizing the embedding $\mathbf{v}$ by solving problem (5) solely target the reconstruction of input images, leading to low text similarity (Gal et al., 2022). Besides, solving problem (5) requires to search in the whole high-dimensional embedding space $\mathbb{E}$, which results in time-consuming training process and slow convergence.

It is natural to construct a textual subspace with high text similarity, in which the searched embedding is able to capture the details of the input image. To this end, vectors with high semantic relevance to $\mathbf{u}$ should be included. As suggested in (Goldberg & Levy, 2014; Le & Mikolov, 2014; Mikolov et al., 2013a;b; Rong, 2014), the vector distance (denoted by $\mathcal{F}$) between the word embeddings in $V$, such as dot product, cosine similarity and $L_2$ norm, can be employed as the semantic similarity of the corresponding words. Now, we are ready to give the distance vector $dist_V$ to calculate the distance between $\mathbf{u}$ and any embedding in $V$, which is given by

$$dist_V = (\mathcal{F}(\mathbf{u}, \mathbf{v}_1), \ldots, \mathcal{F}(\mathbf{u}, \mathbf{v}_{|V|}))^T. \tag{6}$$

Next, we re-order $dist_V$ and the top $M$ vectors are selected as the basis vectors, where at most $d_1 \leq M$ vectors are linearly-independent among them and $d_1$ is the dimension of the textual subspace. The choice of $d_1$ and $\mathcal{F}$ are further discussed in Section 5 and Appendix F. The details of selection strategy are presented in Algorithm 1.

---

**Algorithm 2** BaTex

---

**Input:** image dataset $\mathcal{D}$, pre-trained diffusion network $u_\theta$, reconstruction objective $\mathcal{L}_{rec}$, vector distance $\mathcal{F}$, dimension of textual subspace $d_1$, training iteration $n$, weight decay $\gamma$, vocabulary $V$, initial weights $\{w_0^i\}_{i=1}^M$, initial embedding $\mathbf{u}$, input prompt $S$
Select $M$ embeddings using Algorithm 1: $\mathbf{v}_{i_1}, \ldots, \mathbf{v}_{i_M}$
**Initialize** weights of selected embeddings $\{w_i\}_{i=1}^M \leftarrow \{w_i^0\}_{i=1}^M$
**for** $i = 1$ **to** $n$ **do**
    Sample a mini-batch $\{\boldsymbol{x}\} \sim \mathcal{D}$
    Update embeddings $\{w_i\}_{i=1}^M \leftarrow \nabla_{(w_1,\ldots,w_M)} \mathcal{L}_{rec}(\{\boldsymbol{x}\}, \{w_i\}_{i=1}^M, \gamma)$
**end for**
Get trained weights of the candidate embeddings $\{w_i^*\}_{i=1}^M$
Obtain learned embedding using linear combination $\mathbf{v}^* \leftarrow \sum_{i=1}^M w_i^* \mathbf{v}_{i_M}$
Combine learned embeddings with input prompt $y \leftarrow Combine(S, \mathbf{v}^*)$
Get target images through pretrained diffusion model $\hat{x} \leftarrow u_\theta(y)$
**Output:** target images: $\hat{x}$

---

### 3.3 Concept Generation

Given the chosen embeddings $\mathbf{v}_{i_1}, \ldots, \mathbf{v}_{i_M}$, once the corresponding learned weights $\{w_i^*\}_{i=1}^M$ are obtained by $\arg\min_{\mathbf{v} \in \mathbb{S}} \mathcal{L}_{rec}$ with $\mathbb{S} = \text{span}(\mathbf{v}_{i_1}, \ldots, \mathbf{v}_{i_M})$, the learned embedding $\mathbf{v}^*$ can be formed as:

$$\mathbf{v}^* = w_1^* \mathbf{v}_{i_1} + w_2^* \mathbf{v}_{i_2} + \cdots + w_M^* \mathbf{v}_{i_M}. \tag{7}$$

Then it can be combined with any input textual prompt $S$ as:

$$y \leftarrow Combine(S, \mathbf{v}^*), \tag{8}$$

where the $Combine$ operator is defined in Section 2.3. Subsequently, the target images $\hat{x}$ are generated using the pre-trained diffusion network $u_\theta$:

$$\hat{x} \leftarrow u_\theta(y). \tag{9}$$

Details of BaTex are shown in Algorithm 2.

Finally, we derive that for single-step optimization scenario, the difference of the embedding update between Textual Inversion and BaTex corresponds to a matrix transformation with rank $d_1$, which is the dimension of the textual subspace. The formal theorem is presented as follows, and its proof is included in Appendix C.

**Theorem 2** *For single-step optimization, let $\mathbf{v}_1^* = \mathbf{u} + \Delta\mathbf{v}_1$ and $\mathbf{v}_2^* = \mathbf{u} + \Delta\mathbf{v}_2$ be the updated embedding of Textual Inversion and BaTex respectively, where $\mathbf{u}$ is the initial embedding. Then there exists a matrix $B_V \in \mathbb{R}^{d \times d}$ with rank $d_1$, such that*

$$\Delta\mathbf{v}_2 = B_V \Delta\mathbf{v}_1,$$

*where $d_1$ is the dimension of the textual subspace ($d_1 < d$).*

It can be seen from Theorem 2 that BaTex actually defines a transformation from $\mathbb{R}^d$ to $\mathbb{R}^{d_1}$ using the selection strategy stated in Algorithm 1, which intuitively benefits for optimization process since $B_V$ is formed by the pre-trained embeddings, showing that BaTex explicitly extracts more information from the vocabulary $V$.

## 4 Experiments

### 4.1 Experimental Settings

In this subsection, we present the experimental settings, and more details can be found in Appendix E.

We compare the proposed BaTex with Textual Inversion (TI) (Gal et al., 2022), the original method for personalized generation which lies in the category of "Embedding Optimization". To analyze the

| Category | Method | Param Size | Training Steps | Training Time | High Image-Alignment | High Text-Alignment | Avoid Overfitting |
|---|---|---|---|---|---|---|---|
| Model Optimization | Dreambooth | 860M | 800 | 12min | ✓ | ✗ | ✗ |
| | Custom-Diffusion | 57.1M | 250 | 10min | ✗ | ✓ | ✗ |
| Embedding Optimization | Textual Inversion | 768 | 3000 | 1h | ✓ | ✗ | ✓ |
| | BaTex (Ours) | < 768 | 500 | 10min | ✓ | ✓ | ✓ |

Table 1: A comparison of the abilities of different methods.

Figure 2: Visualization comparison of different methods.

quality of learned embeddings, we follow the most commonly used metrics in TI and measure the performance by computing the CLIP-space scores (Hessel et al., 2021).

We also compare with two "Model Optimization" methods, DreamBooth (DB) (Ruiz et al., 2022) and Custom Diffusion (CD) (Kumari et al., 2022). DB finetunes all the parameters in Diffusion Models, resulting in the ability of mimicing the appearance of subjects in a given reference set and synthesize novel renditions of them in different contexts. However, it finetunes a large amount of model parameters, which leads to overfitting (Ramasesh et al., 2022). CD compares the effect of model parameters and chooses to optimize the parameters in the cross-attention layers. While it provides an efficient method to finetune the model parameters, it requires to prepare a regularized dataset (extracted from LAION-400M dataset (Schuhmann et al., 2021)) to mitigate overfitting, which is time-consuming and hinders its scalability to on-site application. A detailed comparison of method ability can be seen in Table 1.

## 4.2 QUALITATIVE COMPARISON

We first show that learning in a textual subspace significantly improves the text similarity of learned embedding while retaining the ability to reconstruct the input image. The results of text-guided synthesis are shown in Figure 2. As can be seen, for complex input textual prompt with additional text conditions, our method completely captures the input concept and naturally combines it with known concepts.

Additionally, We show the effectiveness of our method by composing two concepts together and introducing additional text conditions (shown in bolded text). The results are shown in Figure 3. It can be seen that BaTex not only allows for the lossless combination of two distinct concepts, but also faithfully generates the additional text conditions.

| Category | Method | Metric | Cat [5] | Wooden-pot [4] | Gta5-artwork [14] | Anders-zorn [12] | Cute-game [8] | **Mean** |
|---|---|---|---|---|---|---|---|---|
| Model Optimization | DB | Text | 0.74 (0.00) | 0.63 (0.01) | 0.72 (0.01) | 0.72 (0.01) | 0.62 (0.00) | 0.69 |
| | | Image | 0.91 (0.01) | 0.88 (0.01) | 0.61 (0.01) | 0.74 (0.00) | 0.61 (0.01) | 0.75 |
| | CD | Text | 0.79 (0.00) | 0.71 (0.00) | 0.74 (0.01) | 0.74 (0.01) | 0.74 (0.00) | 0.74 |
| | | Image | 0.87 (0.00) | 0.75 (0.00) | 0.59 (0.01) | 0.60 (0.02) | 0.56 (0.01) | 0.68 |
| Embedding Optimization | TI | Text | 0.62 (0.00) | 0.66 (0.01) | 0.78 (0.01) | 0.72 (0.01) | 0.72 (0.00) | 0.70 |
| | | Image | 0.89 (0.01) | 0.81 (0.00) | 0.67 (0.01) | 0.67 (0.01) | 0.68 (0.01) | 0.74 |
| | **BaTex** | Text | 0.76 (0.00) | 0.72 (0.01) | 0.80 (0.00) | 0.77 (0.00) | 0.77 (0.01) | 0.76 |
| | | Image | 0.88 (0.01) | 0.81 (0.01) | 0.66 (0.01) | 0.72 (0.00) | 0.66 (0.01) | 0.74 |

Table 2: Quantitative comparison between BaTex and previous works. The numbers in square bracket and parenthesis are the number of image in the dataset and the standard deviation. "Text" and "Image" refer to text-image and image-image alignment scores respectively. Red and blue numbers indicate the best and second best results respectively. While our method achieves similar results to TI in terms of image reconstruction, we significantly outperform them in terms of text similarity, and even achieve results comparable to the methods of the "Model Optimization" category. The results are reported with standard deviation over five random seeds.

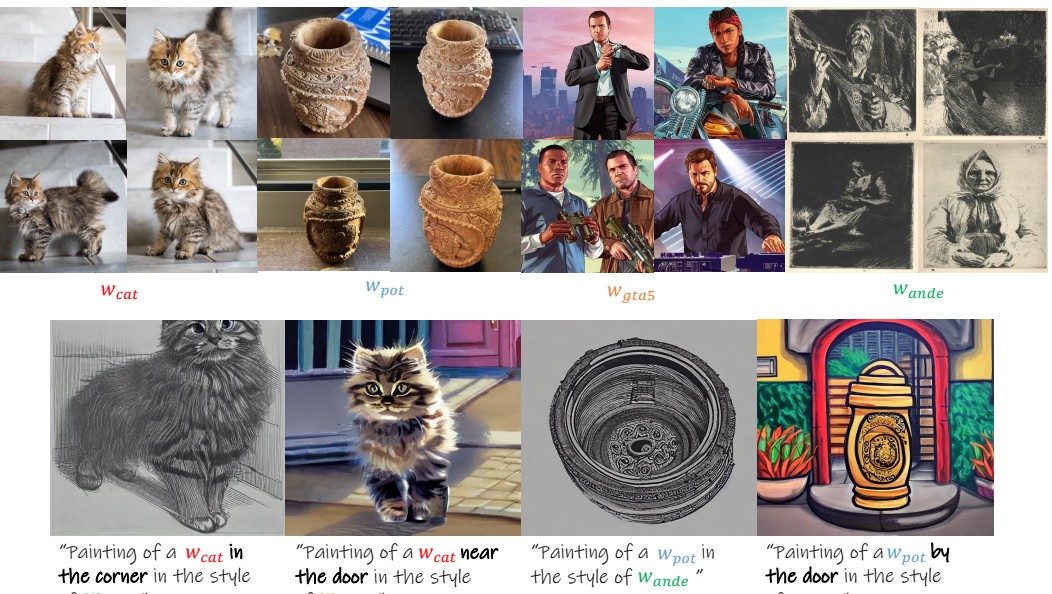

Figure 3: Concept composition of multiple learned embeddings. Bolded texts are additional text conditions.

## 4.3 QUANTITATIVE COMPARISON

The results of image-image and text-image alignment scores compared with previous works are shown in Table 2. As can be seen, when compared with TI by text-image alignment score, BaTex substantially outperforms it (0.76 to 0.70) while maintaining non-degrading image reconstruction effect (0.74 to 0.74). For "Model Optimization" category, BaTex is competitive in both metrics, while their methods perform poorly in one of them due to overfitting. Additional results can be found in Appendix F.

## 4.4 USER STUDY

Following Textual Inversion, we have conducted a human evaluation with two test phases of image-image and text-image alignments. We collected a total of 160 responses to each phase. The results are presented in Table 3, showing that the human evaluation results align with the CLIP scores.

| Metric | DB | CD | TI | Ours |
|---|---|---|---|---|
| Image-to-image alignment | 63.8 | 48.1 | 57.5 | 67.5 |
| Text-to-image alignment | 77.5 | 58.1 | 27.5 | 71.9 |

Table 3: Human preference study.

| Metric | M=96 | $M = 192$ | $M = 384$ | $M = 576$ | $M = 672$ |
|---|---|---|---|---|---|
| Text-image alignment score | 0.77 | 0.78 | 0.80 | 0.80 | 0.79 |
| Image-image alignment score | 0.59 | 0.61 | 0.64 | 0.66 | 0.66 |
| Convergence steps | 150 | 100 | 400 | 500 | 1000 |

Table 4: Results of text-image and image-image alignment scores and convergence steps of dataset *Gta5-artwork* with respect to the dimension of textual subspace.

## 5 DISCUSSION

In this section, we discuss the effects of proposed BaTex. Since the dimension of the textual subspace highly affects the search space of the target embedding, we perform an ablation study on the dimension $M$ and training steps. We also analyze the robustness and flexibility of BaTex by replacing the initial word. The results can be found in Appendix F. The limitations and societal impact of BaTex are discussed in Appendix A and D respectively. The reproducibility statement is presented in Appendix G.

**Dimension of textual subspace** The choice of $M$ is significant to our method since it affects the solution space of target embedding. Specifically, we compare the text-alignment and image-alignment scores by the following numbers: $\{96, 192, 384, 576, 672\}$ (Since $d = 768$, we only compare $M$ values less than 768). We show the results of dataset *Gta5-artwork* with respect to the dimension $M$ in Table 4. As can be seen, choosing $M = 576$ leads to relatively better results. The reasons are two-fold. First, for textual subspace with excessive dimension, optimizing is inefficient and requires more convergence steps as shown in the column of "$M = 672$". Second, For low-dimensional textual subspace, although it generally converges faster, it is difficult to reconstruct the input image as can be seen in the row "Image-image alignment score". We also observe a slight decrease in the value of text-image alignment score as the dimension decreases, which can be explained by the fact that it might not include enough semantic-related embeddings as its basis vectors. Thus, we choose to set $M = 576$ although it is possible to finetune $M$ for each dataset.

**Training steps** The convergence steps of dataset *Gta5-artwork* with respect to the dimension $M$ are shown in Table 4. We observe that when lowering the dimension, it significantly improves the convergence speed. We also notice an outlier for "$M = 96$", which can be explained by its low image-image alignment score, making it difficult to converge. Thus, We recommend to train BaTex with $M = 576$ for 500 steps.

## 6 CONCLUSION

We have proposed BaTex, a novel method for efficiently learning arbitrary concept in a textual subspace. Through a rank-based selection strategy, BaTex determines the textual subspace using the information from the vocabulary, which is time-efficient and better preserves the text similarity of the learned embedding. On the theoretical side, we demonstrate that the selected embeddings can be combined to produce arbitrary vectors in the embedding space and the proposed method is equivalent to applying a projection matrix to the update of embedding. We experimentally demonstrate the efficiency and robustness of the proposed BaTex. Future improvements include introducing sparse optimization algorithm to automatically choose the dimension of textual subspace, and combining with "Model Optimization" methods to improve its image-image alignment score.

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

# Supplementary Materials

## A    LIMITATIONS

Since BaTex requires to form a textual subspace to search the target embedding, the dimension $M$ has to be defined. In BaTex , $M$ has been treated as a hyperparameter and we have performed a detailed comparison in Section 5 to choose an appropriate value. However, in practice, $M$ should be determined automatically to avoid artificially tuning. In Section 6, we have given some directions for improvement to address this issue.

## B    PROOF OF THEOREM 1

**Theorem 3** *Any vector $\mathbf{v}$ in word embedding space $\mathbb{E}$ can be represented by a linear combination of the embeddings in vocabulary $V$.*

**Proof 1** *For all embeddings $\mathbf{v}_1, \ldots, \mathbf{v}_{|V|} \in \mathbb{E}$ in $V$, it forms a matrix $A_V = [\mathbf{v}_1, \ldots, \mathbf{v}_{|V|}] \in \mathbb{R}^{d \times |V|}$. It can be numerically computed that the rank of $A_V$ is $r(A_V) = d$, demonstrating that any $d$ linearly-independent vectors in $V$ can be formed as a basis of the word embedding space $\mathbb{E}$. Since $\mathbb{E} = \mathbb{R}^d$, we can derive that if we select a subset of $d$ linearly-independent vectors $(\mathbf{v}_{i_1}, \ldots \mathbf{v}_{i_d})$ from $V$, it forms a basis of $\mathbb{E}$. Then any vector $\mathbf{v} \in \mathbb{E}$ can be expressed as a linear combination of the subset $(v_{i_1}, \ldots v_{i_d})$ as: $\mathbf{v} = w_{i_1}\mathbf{v}_{i_1} + \cdots + w_{i_d}\mathbf{v}_{i_d}$, where $w_{i_1}, \ldots, w_{i_d}$ are the weights of the basis vectors.*

## C    PROOF OF THEOREM 2

**Theorem 4** *For single-step optimization, let $\mathbf{v}_1^* = \mathbf{u} + \Delta\mathbf{v}_1$ and $\mathbf{v}_2^* = \mathbf{u} + \Delta\mathbf{v}_2$ be the updated embedding of Textual Inversion and BaTex respectively, where $\mathbf{u}$ is the initial embedding. Then there exists a matrix $B_V \in \mathbb{R}^{d \times d}$ with rank $d_1$, such that $\Delta\mathbf{v}_2 = B_V\Delta\mathbf{v}_1$, where $d_1$ is the dimension of the textual subspace ($d_1 < d$).*

**Proof 2** *For BaTex , we have $\mathbf{u} = V_M\mathbf{w}^0$, where $V_M \in \mathbb{R}^{d \times M}$ and $\mathbf{w}^0 \in \mathbb{R}^M$ are the selected embedding matrix and initialized weights respectively, $M$ is the number of embeddings selected. We also have $r(V_M) = d_1$. For single-step optimization, let $\mathbf{w}^* = \mathbf{w}^0 + \Delta\mathbf{w}$ be the updated weights, then $\Delta\mathbf{w} = \nabla_{\mathbf{w}}\mathcal{L}_{rec}$, where $\mathcal{L}_{rec}$ is the objective of the reconstruction task. Now we have $\mathbf{v}_2^* = V_M\mathbf{w}^* = V_M(\mathbf{w}^0 + \Delta\mathbf{w}) = \mathbf{u} + \Delta\mathbf{v}_2$. So $\Delta\mathbf{v}_2 = V_M\nabla_{\mathbf{w}}\mathcal{L}_{rec}$. Using the chain rule, we know that $\nabla_{\mathbf{w}}\mathcal{L}_{rec} = V_M^T\nabla_{\mathbf{v}}\mathcal{L}_{rec}$, which draws a conclusion that $\Delta\mathbf{v}_2 = V_M V_M^T\nabla_{\mathbf{v}}\mathcal{L}_{rec}$. Also, we know that for single-step optimization, $\Delta\mathbf{v}_1 = \nabla_{\mathbf{v}}\mathcal{L}_{rec}$ (Here we omit the learning rate and other hyperparameters). Now we have $\Delta\mathbf{v}_2 = V_M V_M^T\Delta\mathbf{v}_1$. Set $B_V = V_M V_M^T$, we have $r(B_V) = r(V_M V_M^T) = r(V_M) = d_1$.*

## D    SOCIETAL IMPACT

With the gradual increase in the capacity of multimodal models in recent years, training a large model from scratch is no longer possible for most people. Our method allows users to combine private images with arbitrary text for customized generation. Our method is time and parameter efficient, allowing users to take advantage of a large number of pre-trained parameters, promising to increase social productivity in image generation. While being expressive and efficient, it might increase the potential danger in misusing it to generate fake or illegal data. Possible solutions include enhancing the detection ability of diffusion model (Corvi et al., 2022) and constructing safer vocabulary of Text model (Devlin et al., 2018).

## E    ADDITIONAL EXPERIMENTAL SETTINGS

**Datasets**    Following the existing experimental settings, we conduct experiments on several concept datasets, including datasets from Textual Inversion (Gal et al., 2022) and Custom Diffusion (Kumari

et al., 2022). The datasets are all publicly available by the authors, as can be seen in the website of Textual Inversion[6] and Custom-Diffusion[7]. Besides, we collect several complex concept datasets from HuggingFace Library[8] and perform detailed comparisons on them. It is worth noting that none of the datasets we have used contain personally identifiable information or offensive content.

**Baselines** We compare our method with Textual Inversion (Gal et al., 2022), the original method for concept generation which lies in the category of "Embedding Optimization". We also compare with two "Model Optimization" methods, DreamBooth (Ruiz et al., 2022) and Custom Diffusion (Kumari et al., 2022). DreamBooth finetunes all the parameters in Diffusion Models, resulting in the ability of mimicing the appearance of subjects in a given reference set and synthesize novel renditions of them in different contexts. However, it finetunes a large amount of model parameters, which leads to overfitting (Ramasesh et al., 2022). Custom-Diffusion compares the effect of model parameters and chooses to optimize the parameters in the cross-attention layers. While it provides an efficient method to finetune the model parameters, it requires to prepare a regularized dataset (extracted from LAION-400M dataset (Schuhmann et al., 2021)) to mitigate overfitting, which is time-consuming and hinders its scalability to on-site application. A detailed comparison of method ability can be seen in Table 1 in the main text.

**Implementation** We implement all three baseline methods by Diffusers[9], a third-party implementation compatible with Stable Diffusion (Rombach et al., 2022a), which is more expressive than the original Latent Diffusion Model. All four models are used with Stable-Diffusion-v1-5[10].

**Evaluation Metrics** To analyze the quality of learned embeddings, we follow the most commonly used metrics in Textual Inversion and measure the performance by computing the clip-space scores (Hessel et al., 2021). We use the original implementation of CLIPScore[11].

We first evaluate our proposed method on image alignment, as we wish our learned embeddings still retain the ability to reconstruct the target concept. For each concept, we generate 64 examples using the learned embeddings with the prompt "A photo of *". We then compute the average pair-wise CLIP-space cosine similarity between the generated images and the images in the training dataset.

Secondly, we want to measure the alignment of the generated images with the input textual prompt. We have set up a series of prompts and included as many variations as possible. These include background modifications ("A photo of * on the beach") and style transfer ("a waterfall in the style of *").

**Hardware** Our method and all previous works are trained on three NVIDIA A100 GPUs. For inference, BaTex uses a single NVIDIA A100 GPU.

**Hyperparameter settings** We optimize the weights using AdamW (Loshchilov & Hutter, 2017), the same optimizer as Textual Inversion. The choice of other important hyperparameters are all discussed in Section 5.

## F ADDITIONAL RESULTS

### F.1 QUANTITATIVE COMPARISON

We perform quantitative comparison on five additional datasets. Results are shown in Table 5. As can be seen, when compared with TI by text-image alignment score, BaTex substantially outperforms it (0.75 to 0.66) while maintaining non-degrading image reconstruction effect (0.76 to 0.76). For "Model Optimization" category, BaTex is competitive in both metrics, while their methods perform poorly in one of them due to overfitting.

---

[6]https://github.com/rinongal/textual_inversion

[7]https://github.com/adobe-research/custom-diffusion

[8]https://huggingface.co/sd-concepts-library

[9]https://github.com/huggingface/diffusers

[10]https://huggingface.co/runwayml/stable-diffusion-v1-5

[11]https://github.com/jmhessel/clipscore

| Category | Method | Metric | Low-poly [9] | Midjourney-style [4] | Chair [4] | Dog [8] | Elephant [5] | **Mean** |
|---|---|---|---|---|---|---|---|---|
| Model Optimization | DB | Text | 0.74 (0.00) | 0.68 (0.01) | 0.63 (0.01) | 0.74 (0.01) | 0.64 (0.01) | 0.69 |
| | | Image | 0.69 (0.01) | 0.66 (0.01) | 0.91 (0.01) | 0.79 (0.00) | 0.89 (0.01) | 0.79 |
| | CD | Text | 0.75 (0.00) | 0.74 (0.01) | 0.75 (0.01) | 0.73 (0.01) | 0.68 (0.00) | 0.73 |
| | | Image | 0.66 (0.01) | 0.60 (0.01) | 0.82 (0.00) | 0.73 (0.01) | 0.80 (0.00) | 0.72 |
| Embedding Optimization | TI | Text | 0.73 (0.01) | 0.68 (0.00) | 0.58 (0.00) | 0.65 (0.00) | 0.68 (0.01) | 0.66 |
| | | Image | 0.70 (0.01) | 0.66 (0.01) | 0.88 (0.00) | 0.76 (0.00) | 0.84 (0.01) | 0.76 |
| | **BaTex** | Text | 0.77 (0.00) | 0.74 (0.00) | 0.70 (0.01) | 0.74 (0.01) | 0.79 (0.00) | 0.75 |
| | | Image | 0.70 (0.01) | 0.70 (0.00) | 0.88 (0.01) | 0.76 (0.01) | 0.80 (0.00) | 0.76 |

Table 5: Quantitative comparison between BaTex and previous works. The numbers in square bracket and parenthesis are the number of image in the dataset and the standard deviation. "Text" and "Image" refer to text-image and image-image alignment scores respectively. Red and blue numbers indicate the best and second best results respectively. While our method achieves similar results to TI in terms of image reconstruction, we significantly outperform them in terms of text similarity, and even achieve results comparable to the methods of the "Model Optimization" category. The results are reported with standard deviation over five random seeds.

| **Method** | **Text Alignment** | | **Image Alignment** | |
|---|---|---|---|---|
| | "cat" | "animal" | "cat" | "animal" |
| Dreambooth | 0.74 | 0.63 | 0.91 | 0.91 |
| Custom-Diffusion | 0.79 | 0.69 | 0.87 | 0.82 |
| Textual Inversion | 0.62 | 0.62 | 0.89 | 0.86 |
| BaTex (ours) | 0.76 | 0.75 | 0.88 | 0.88 |

Table 6: Comparison of BaTex and three previous works about the robustness against initialization word. We take the *Cat* dataset as an example and use "animal" to replace the original initialization word "cat".

## F.2 ROBUSTNESS AGAINST INITIAL WORD

In practice, it is often difficult for users to give the most suitable initial word for complex concepts at once, posing a great challenge to the training of previous models. We show in Table 6 that when initializing by a less semantic-related word, BaTex outperforms all three previous models by a huge margin in terms of text-image alignment score. The robustness comes from the fact that BaTex selects $M$ embeddings by the initial word, which contains almost all synonyms or semantically similar words of the initial word.

We additionally test the robustness against initial word on *Dog* dataset and replace the initial word "dog" by "animal". Results are shown in Table 7. It can be seen that when the text similarity of other three previous methods significantly decreases, BaTex obtains a much higher score.

## F.3 TEXT-GUIDED SYNTHESIS

We show additional text-guided synthesis results on Figure 4 and 5. As can be seen, for complex input textual prompt with additional text conditions, our method completely captures the input concept and naturally combines it with known concepts.

Also, qualitative results of more object concepts are shown in Figure 6. The results match the conclusion in Section 4.2.

## F.4 VECTOR DISTANCE

We compare the text-image alignment score of three type of vector distance: dot product, cosine similarity and $L_2$ distance in Figure 7. As can be seen, dot product and cosine similarity both outperform $L_2$ in text-image alignment score and convergence speed. Compared with cosine similarity, the results of dot product are slightly better. We conjecture that the reason is dot product taking

| Method | Text Alignment | | Image Alignment | |
|---|---|---|---|---|
| | "dog" | "animal" | "dog" | "animal" |
| Dreambooth | 0.73 | 0.65 | 0.79 | 0.78 |
| Custom-Diffusion | 0.74 | 0.65 | 0.74 | 0.77 |
| Textual Inversion | 0.65 | 0.60 | 0.76 | 0.76 |
| BaTex (ours) | 0.73 | 0.73 | 0.75 | 0.74 |

Table 7: Comparison of BaTex and three previous works about the robustness against initialization word. We take the *Dog* dataset as an example and use "animal" to replace the original initialization word "dog".

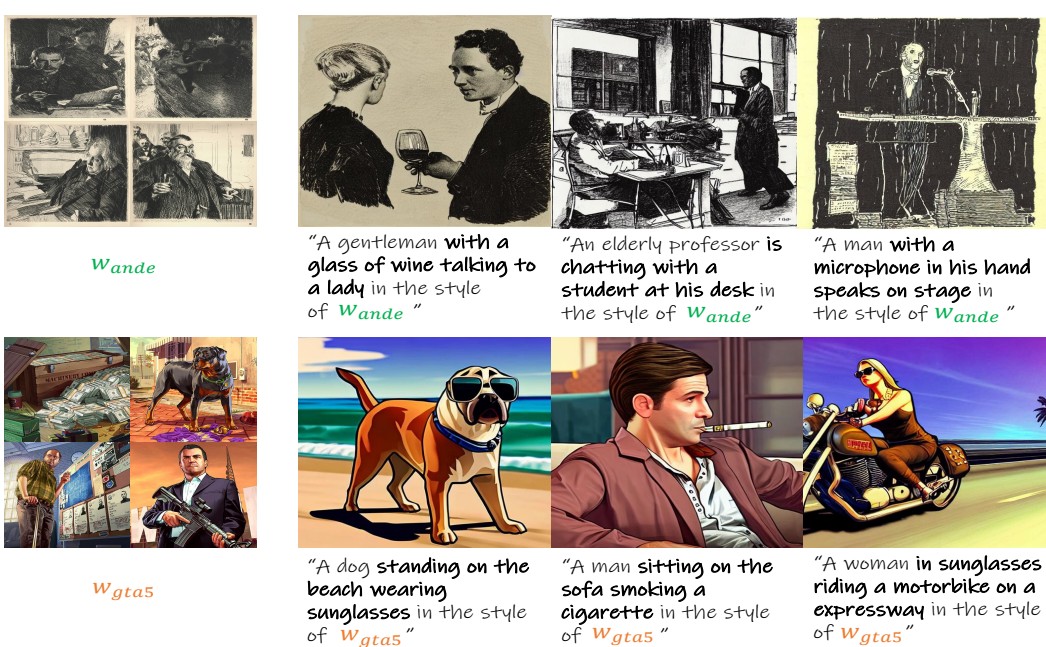

Figure 4: Qualitative results of BaTex. Bolded texts are additional text conditions.

into account the modal length of the embedding when computing the distance between **u** and other embeddings, which is meaningful in the textual subspace.

## F.5 TOP ONE VECTOR

Simply using the top one vector would lead to worse reconstruction results since only a scalar cannot fully store the details of the reference images. We have also added qualitative results using the top one vector in Figure 8, showing that using only one vector results in unsatisfying results.

## F.6 TWO SIMILAR OBJECTS

We have performed additional multi-concept generation for similar objects and the results are shown in Figure 9. The results show the appearance of object neglect (in the right-most column). We also find that BaTex can generate reasonable images for most cases of two similar objects (see the results of other three columns).

## F.7 TWO DIFFERENT OBJECTS

In addition, we have provided additional experiments about composition of different objects in Figure 10, showing that BaTex successfully composites different objects.

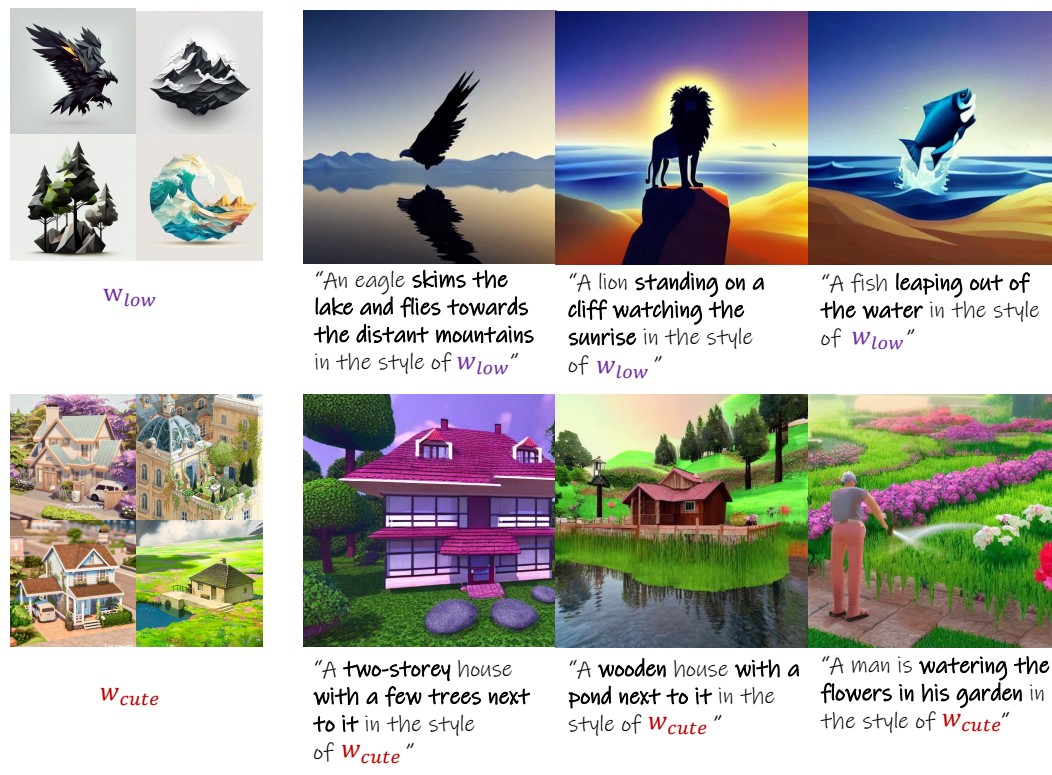

Figure 5: Additional qualitative results of BaTex. Bolded texts are additional text conditions.

| Method | Metric | Cat [5] | Wooden-pot [4] | Gta5-artwork [14] | Anders-zorn [12] | Cute-game [8] | **Mean** |
|--------|--------|---------|----------------|-------------------|------------------|---------------|----------|
| XTI | Text | 0.78 (0.01) | 0.73 (0.01) | 0.67 (0.01) | 0.73 (0.00) | 0.67 (0.00) | 0.72 |
|     | Image | 0.85 (0.00) | 0.74 (0.01) | 0.54 (0.01) | 0.63 (0.01) | 0.56 (0.01) | 0.66 |
| NeTI | Text | 0.76 (0.00) | 0.72 (0.01) | 0.74 (0.00) | 0.71 (0.01) | 0.72 (0.00) | 0.73 |
|      | Image | 0.87 (0.01) | 0.76 (0.00) | 0.64 (0.01) | 0.69 (0.00) | 0.64 (0.01) | 0.72 |
| SVDiff | Text | 0.79 (0.00) | 0.71 (0.00) | 0.76 (0.01) | 0.74 (0.01) | 0.76 (0.01) | 0.75 |
|        | Image | 0.85 (0.01) | 0.75 (0.00) | 0.58 (0.01) | 0.56 (0.01) | 0.59 (0.01) | 0.67 |
| **BaTex** | Text | 0.76 (0.00) | 0.72 (0.01) | 0.80 (0.00) | 0.77 (0.00) | 0.77 (0.01) | 0.76 |
|           | Image | 0.88 (0.01) | 0.81 (0.01) | 0.66 (0.01) | 0.72 (0.00) | 0.66 (0.01) | 0.74 |

Table 8: Additional quantitative comparison between BaTex and concurrent works. The numbers in square bracket and parenthesis are the number of image in the dataset and the standard deviation. "Text" and "Image" refer to text-image and image-image alignment scores respectively. Red and blue numbers indicate the best and second best results respectively.

## F.8 Number Comparison

A comparison about different numbers (N = 1, 2, 4, 6, 8) of reference images over dataset Dog (the maximum number of reference images is 8) is shown in Figure 11. The experiments show that using only 2 images can obtain satisfying results. Although the performance of N = 1 is slightly deteriorating, BaTex mainly focuses on improving the training efficiency and text alignment over Textual Inversion.

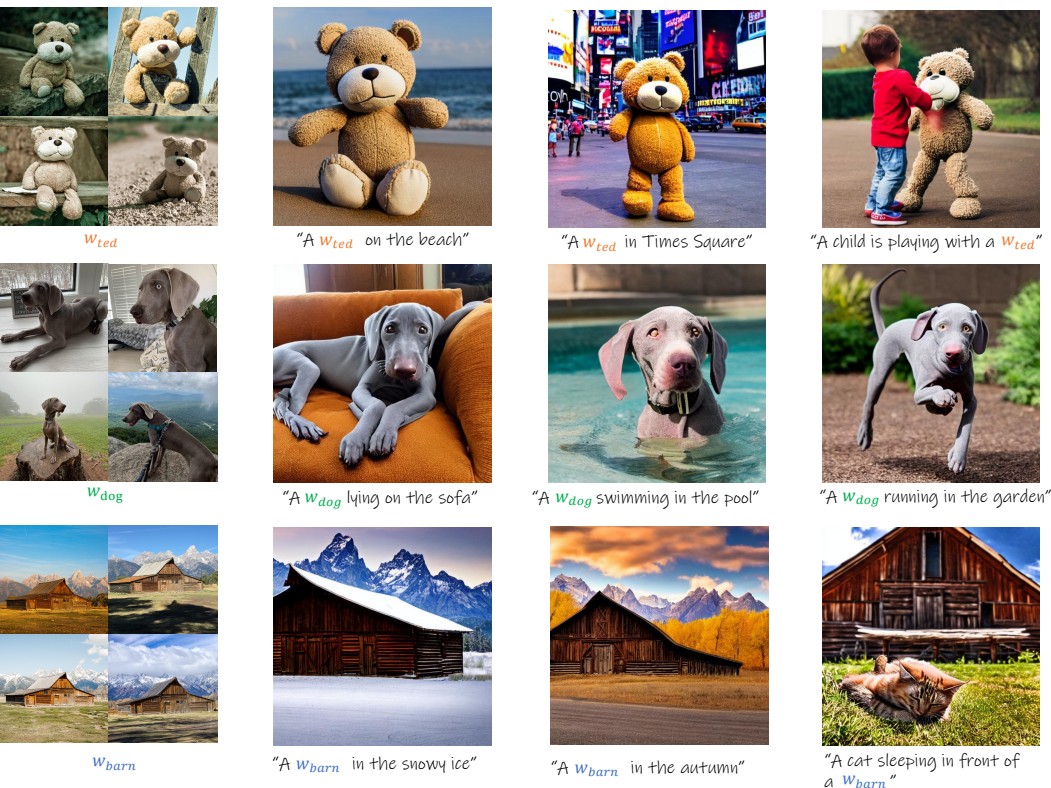

Figure 6: Qualitative results of object concept.

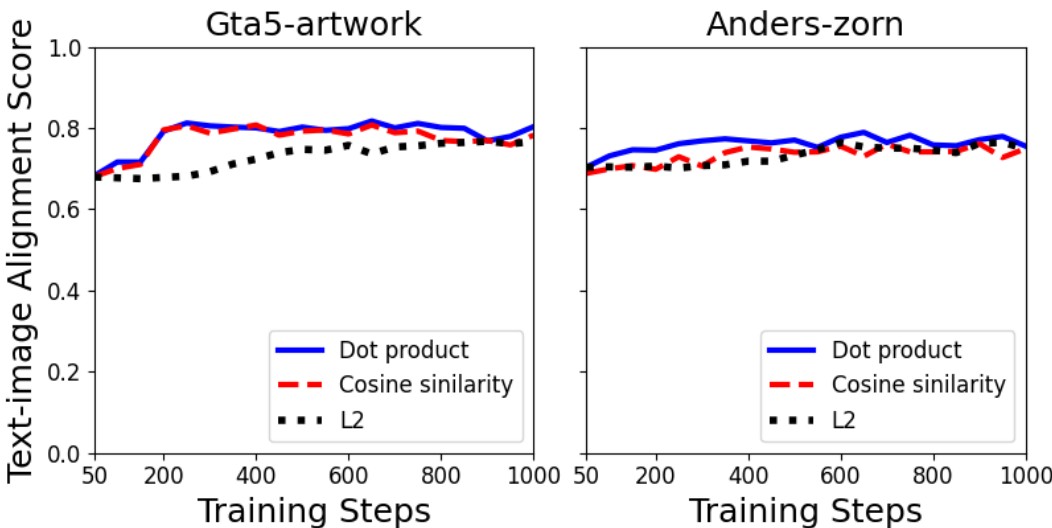

Figure 7: Comparison about text-image alignment score of different distance measurement with respect to training steps.

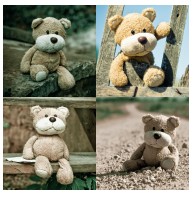 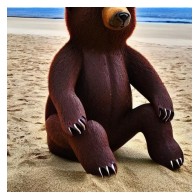 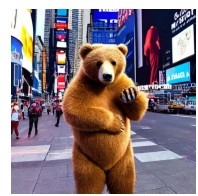 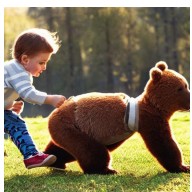

$w_{ted}$   "A $w_{ted}$ on the beach"   "A $w_{ted}$ in Times Square"   "A child is playing with a $w_{ted}$"

Figure 8: Qualitative results using the top one vector.

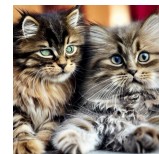 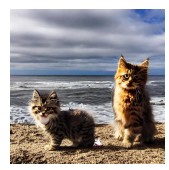 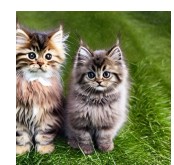 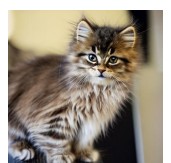

"A $w_1$ lying down with a $w_2$"   "A $w_1$ and a $w_2$ standing on the beach"   "A $w_1$ and a $w_2$ sitting on the grass"   "A $w_1$ looking at a $w_2$"

Figure 9: Qualitative results using two similar concepts.

### F.9 ADDITIONAL COMPARISON

We have also compared BaTex with three concurrent works: SVDiff[12] (Han et al., 2023), XTI[13] (Voynov et al., 2023) and NeTI[14] (Alaluf et al., 2023). Quantitative results are shown in Table 8. It can be seen that BaTex achieves superior results over all three concurrent works. Also, we have shown qualitative comparison results in Figure 12, which demonstrates the effectiveness of our method over state-of-the-art methods.

Also, We perform qualitative comparison with encoder-based method (Wei et al., 2023) using example "cat". Results have been shown in Figure 13. It can be seen that although encoder-based method finetunes much faster (it only needs one forward step), it lacks text-to-image alignment ability in some cases (e.g., miss "sleeping" in the second case).

### F.10 WEIGHT VISUALIZATION

To clarify the usage of basis vectors, we have shown the visualization of both learned weights and corresponding basis vectors for example "cat" in Figure 14. Due to page limit, we only showcase the first 10 basis vectors. As can be seen, starting from the initial word "cat", BaTex learns a reasonable combination to obtain the target vector while TI tends to search around the initial embedding using only the reconstruction loss, which results in low text-to-image alignment.

### F.11 HUMAN FACE RESULTS

Human face domain is very challenging for personalization of image diffusion models, which contains more perceptible details than other domains. Therefore, we have performed test on "lecun" example used in (Gal et al., 2023). Results have been shown in Figure 15. It can be seen that BaTex generates results following the textual prompt, while successfully reconstructing the input human face image.

## G REPRODUCIBILITY STATEMENT

We have provided main part of our code in the supplementary material. Please refer there for details of our method.

---

[12]https://github.com/mkshing/svdiff-pytorch

[13]https://github.com/mkshing/prompt-plus-pytorch

[14]https://github.com/NeuralTextualInversion/NeTI

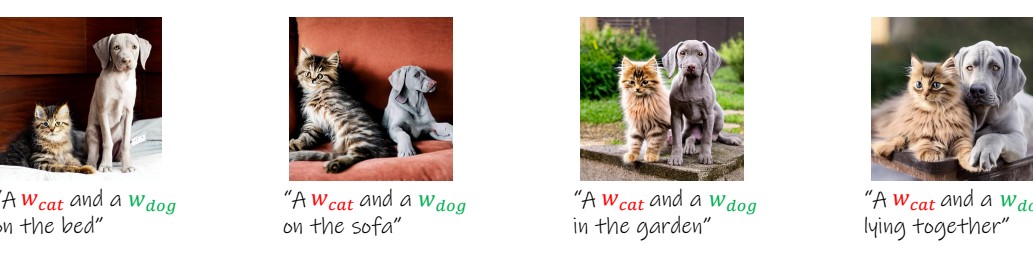

Figure 10: Qualitative results of object composition.

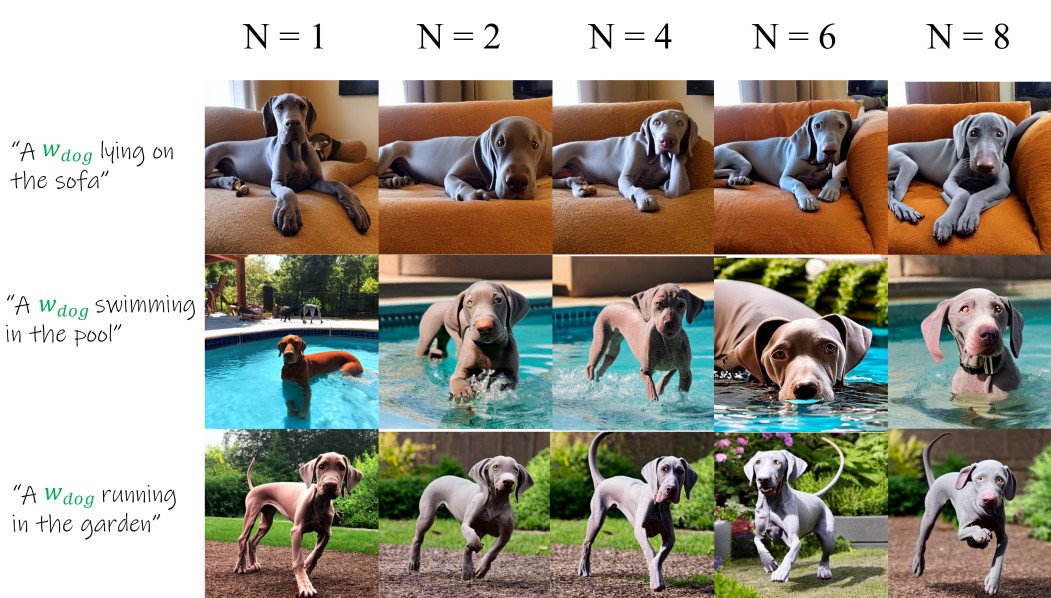

Figure 11: Qualitative comparison of the number of reference images using the *Dog* dataset.

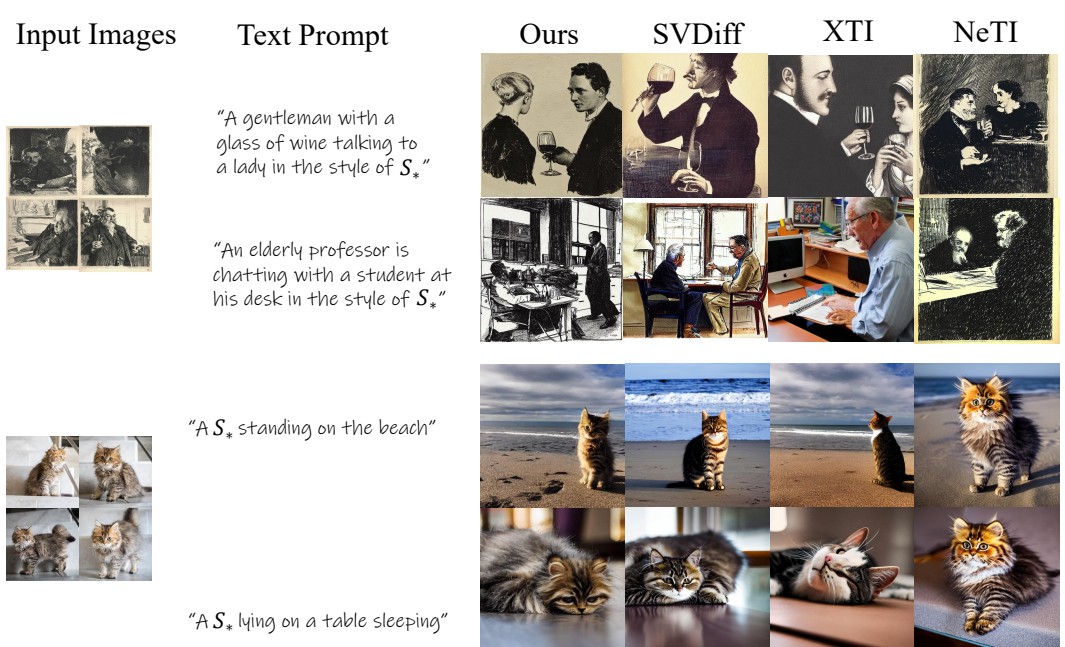

Figure 12: Additional qualitative comparison between BaTex and three concurrent works.

Input Images          Text Prompt                    Ours          ELITE

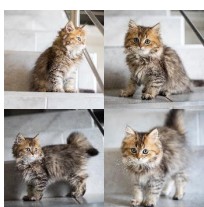

"A $S_*$ standing on the beach"

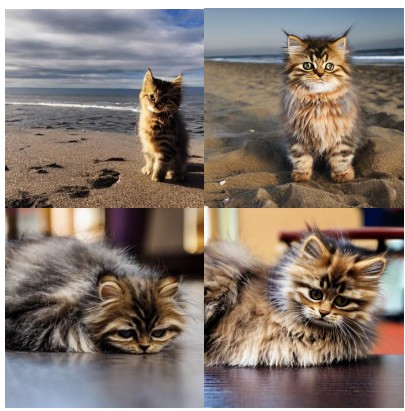

"A $S_*$ lying on a table sleeping"

Figure 13: Qualitative comparison with encoder-based method using example "cat".

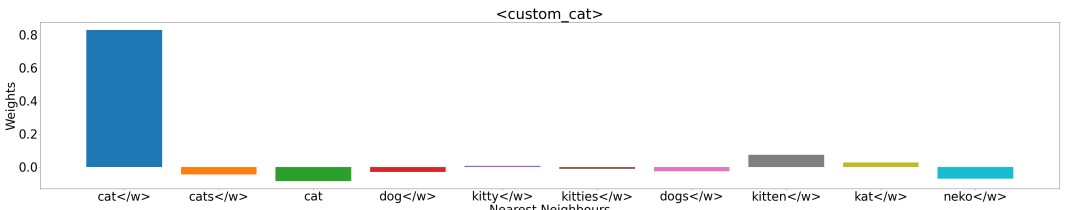

Figure 14: Weight visualization of learned weights and their corresponding basis vectors for example "cat".

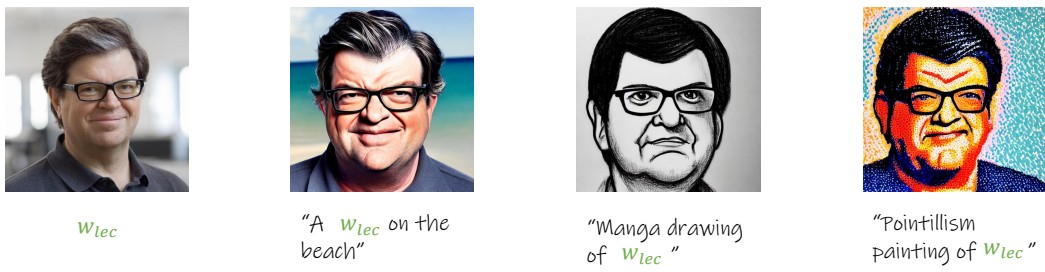

$w_{lec}$          "A $w_{lec}$ on the beach"          "Manga drawing of $w_{lec}$"          "Pointillism painting of $w_{lec}$"

Figure 15: Human face results of "lecun" example.

