# OpenReview forum: "Efficient Personalized Text-to-image Generation by Leveraging Textual Subspace"
_ICLR.cc/2024/Conference — Submitted to ICLR 2024_

### Official Review · Reviewer_uaaP · 2023-11-01

**Soundness:** 3 good
**Presentation:** 2 fair
**Contribution:** 3 good
**Rating:** 6
**Confidence:** 5

**Summary:**

In order to leverage a combined textual prompt and alleviate the computational demands of working in a high-dimensional embedding space, this paper introduces the BaTex method for acquiring versatile embeddings within a low-dimensional textual subspace for personalized text-to-image generation. The experimental results affirm the effectiveness of this approach.

**Strengths:**

1. The paper is excellently structured and offers a clear, comprehensible narrative.

2. This paper presents a novel approach, introducing textual subspace learning to eliminate the need for time-consuming training in high-dimensional embedding spaces.

3. The experiments provide compelling evidence of the efficiency and robustness of the proposed BaTex method.

**Weaknesses:**

1. Regarding the use of subspace learning:

Since the vocabulary V corresponds to a set of pre-trained vectors {v_i}, it might seem logical to directly select the top vector. So, this arises an important problem: why do we need to use the subspace learning method? Firstly, one important motivation is the combination with different textual prompts. I guess that this combination is implemented in a single subspace (and if my understanding is incorrect, kindly correct me). In Figure 2, it is not clear which textual prompts are amalgamated in this single subspace. Additionally, as depicted in Figure 3, the model considers multiple embeddings. Is the model simultaneously exploring multiple subspaces? Secondly, when we compare Figure 6 with Figure 8, we notice that both scenarios using the top vector manage to fulfill the textual descriptions. However, why does Figure 8 fall short in delivering the desired results?

2. Addressing missing objects in generated images:

Figure 2 highlights that the generated images miss some objects mentioned using the TI method. For instance, when examining the images from top to bottom, we notice the absence of objects such as a lady, a student, the beach, and a table. It's important to note that this work primarily focuses on learning a new embedding. How does this learned embedding enable the generation of missing objects? What is the mechanism for achieving this?

3. Clarification on optimization problem and loss:

In Equation (5), the optimization problem is to optimize the variable v. However, in Algorithm 2, the L\_res loss is calculated concerning the variable w. Could you please provide clarification on this?

4. Discussion of limitations:

It appears that the paper lacks a discussion on its limitations. It would be valuable to address and discuss any limitations of this work. What potential constraints or drawbacks should readers be aware of when considering the findings and applications of this research?

**Questions:**

Please see the weaknesses.

---

> ### Author Response · Authors · 2023-11-19
> **Response to Reviewer uaaP**
>
> Q1: Regarding the use of subspace learning...
>
> R1: In the reverse process of diffusion models, to obtain the text condition $y$, all word vectors need to be combined and input through a pre-trained text encoder. In the literature of NLP, all word vectors are assumed to lie in a shared space. One of the motivations of our method is to construct a textual subspace according to the provided initial word and optimize the target vector on it. For multi-concept generation, several trained vectors are combined with other word vectors and input through the text encoder, with each trained vector is searched and optimized in a independent textual subspace. However, although multiple vectors are represented in their textual subspace, they do not need to be trained at the same time. For Figure 6, we have used $M$ basis vectors to represent the object to showcase more qualitative results. However in Figure 8, we only use $1$ basis vector to perform an ablation study to demonstrate the necessity to utilize multiple basis vectors to represent the target concept.
>
> Q2: Addressing missing objects in generated images...
>
> R2: It is widely stated in the literature of text-to-image personalization that TI falls short of text-to-image alignment ability since it only optimizes the word vector in a reconstructive manner, while the proposed method, successfully improves the text-to-image alignment score with low computational cost by constraining it in a textual subspace. The ability of generating missing objects can be contributed to the high text-to-image alignment ability of our method, which successfully reflects all text information.
>
> Q3: Clarification on optimization problem and loss...
>
> R3: From Eq. (7), the target word vector $v$ is represented by the weights $w$ and its corresponding basis vectors. By switching the optimization target from $v$ to $w$, the proposed method represents and optimizes the target vector in a textual subspace, which offers high text-to-image alignment and low computational cost.
>
> Q4: Discussion of limitations...
>
> R4: Due to page limit, the limitations have been discussed in Appendix A. Please head there for details.

---

### Official Review · Reviewer_GqUH · 2023-11-01

**Soundness:** 3 good
**Presentation:** 3 good
**Contribution:** 2 fair
**Rating:** 6
**Confidence:** 2

**Summary:**

This work primarily focuses on personalizing text-to-image models. The authors introduce Batex, a method that leverages multiple text embeddings with high similarity and updates their weights. This approach offers the advantages of reduced training time and improved text-image alignment. The paper provides both qualitative and quantitative results to support the effectiveness of the proposed method.

**Strengths:**

1. The organization and writing of the paper are commendable, resulting in a clear and easily understandable presentation.
2. The proposed method is both simple and effective, as exemplified by the compelling results presented in Table 2.

**Weaknesses:**

1. The authors say that their method offers advantages in terms of reduced training time and improved text-image alignment. However, the significance of improving training time may be less important considering the already inexpensive nature of personalization (text inversion). Additionally, the reasoning behind achieving higher text-image alignment compared to traditional text-to-image (TI) methods is not adequately clarified. Why TI cannot learn an embedding to align text and images?
2. It is hard to tell if Batex outperforms TI in Figure 2. It would be beneficial for the authors to provide further explanations regarding the superiority of Batex in Figure 2 to help readers better understand the comparative strengths of the proposed method.

**Questions:**

1. Pls see weakness.
2. How is the results of other baselines in fig 3?

**Details Of Ethics Concerns:**

N/A.

---

> ### Author Response · Authors · 2023-11-19
> **Response to Reviewer GqUH**
>
> Q1: The significance of improving training time may be less important considering the already inexpensive nature of personalization...
>
> R1: In the literature of text-to-image personalization, TI is often considered to be slow since it requires nearly an hour to train a single object, thus hindering it to be applied to large-scale applications. We have experimentally demonstrated that by applying the proposed BaTex method, the training budget can be reduced to nearly 10 minutes, which is only 1/6 of TI. Also, we have stated in Section 3.2 that the textual subspace is designed using the input word and the vocabulary provided by the pre-trained models. Thus, when optimizing the target vector in this subspace, the text-to-image alignment can be well preserved, which results in high text-to-image alignment score. Conversely, the objective of TI solely depends on the reconstruction loss, which only concentrates on the input image. In a word, our method demonsrates that, by constraining the training process in a textual subspace induced by the vocabulary, the learned vector retains high text-to-image alignment score with less computational cost.
>
> Q2: It is hard to tell if Batex outperforms TI in Figure 2...
>
> R2: Take the upper figures as an example. We have added "with a glass of wine talking to a lady'' and "chatting with a student at his desk'' as additional text conditions. It can be seen that only the results of BaTex successfully reflect the corresponding text conditions while other three methods all fail to achieve it. Although TI does follow the provided style, the results are missing various degree of text information due to its low text-to-image alignment ability.
>
> Q3: How is the results of other baselines in fig 3?
>
> R3: In this paper, we focus on learning a single concept by a word vector in a induced textual subspace to improve its text-to-image alignment with less computational overhead. Thus, we only showcase multi-concept results as a complement to demonstrate the performance of our method. We leave investigation of more detailed multi-concept generation to future study.

---

### Official Review · Reviewer_pRpB · 2023-11-03

**Soundness:** 2 fair
**Presentation:** 3 good
**Contribution:** 2 fair
**Rating:** 6
**Confidence:** 4

**Summary:**

This paper aims to address the issues of prompt editing degradation and time-consuming training process in personalized text-to-image generation.
To this end, it introduces an efficient method to explore the target embedding in a textual subspace with higher text similarity.
Specifically, it proposes a selection strategy to determine the basis vecors of textual subspace.
Experiments demonstrate that the learned embedding can both reconstruct input image and improve its alignment with editing prompts.

**Strengths:**

See summary.

**Weaknesses:**

1. The optimization of target embedding is performed in an explainable textual subspace. So could the authors provide visualizations of both learned weights and corresponding basis vectors (i.e., words)?
2. Too few methods are compared in this paper. The authors are encouraged to add more baselines including optimization-based [1] and encoder-based[2,3] in revision.
3. The expressiveness of words combination may be limited, especically in reconstructing image detalis, e.g., human faces.
4. The selection stratgy of textual subspace chooses M basis embeddings that are most similar to initialization embedding u. Is there too much redundancy among these basis embeddings?

[1] Ligong Han, et al. SVDiff: Compact Parameter Space for Diffusion Fine-Tuning. ICCV 2023.

[2] Yuxiang Wei, et al. Elite: Encoding visual concepts into textual embeddings for customized text-to-image generation. ICCV 2023.

[3] Rinon Gal, et al. Encoder-based Domain Tuning for Fast Personalization of Text-to-Image Models. SIGGRAPH 2023.

**Questions:**

See weaknesses.

---

> ### Author Response · Authors · 2023-11-19
> **Response to Reviewer pRpB**
>
> Q1: Could the authors provide visualizations of both learned weights and corresponding basis vectors (i.e., words)?
>
> R1: We have provided the learned weights and their corresponding basis vectors for example ``cat'' in Appendix F.10. As can be seen, starting from the initial word “cat”, BaTex learns a reasonable combination to obtain the target vector while TI tends to search around the initial embedding using only the reconstruction loss, which results in low text-to-image alignment score. Please head there for more details.
>
> Q2: Too few methods are compared in this paper...
>
> R2: Thanks for advice. We have added both qualitative and quantitative comparisons with three concurrent works: SVDiff [1], XTI [2] and NeTI [3]. For encoder-based methods mentioned above, since they require to train a domain encoder for each object category, it is not suitable for most cases in our experiments. We will try to compare with these methods in the next revision. Results can be seen in Appendix F.9. It has shown that our method achieves superior results over all three concurrent works. Please head there for details.
>
> Q3: The expressiveness of words combination may be limited, especically in reconstructing image detalis, e.g., human faces.
>
> R3: We have demonstrated the expressiveness of our method from qualitative and quantitative comparisons. As can be seen in Table 2, our method achieves superior text-to-image alignment score and comparable image-to-image alignment score compared to baseline methods. We have also included additional qualitative and quantitative results in Appendix F, including object composition and more qualitative comparisons.
>
> Q4: Is there too much redundancy among these basis embeddings?
>
> R4: We have conducted an ablation study about the number $M$ using the text-to-image alignment score and image-to-image alignment score in Table 4 in Section 5. It can be seen that setting $M=576$ achieves a balance between high generation quality and fast convergence speed. We leave further analysis of $M$ to future study.
>
> [1] Ligong Han, et al. SVDiff: Compact Parameter Space for Diffusion Fine-Tuning. ICCV 2023.
>
> [2] Voynov, et al. P+: Extended Textual Conditioning in Text-to-Image Generation. arXiv:2303.09522.
>
> [3] Alaluf, et al. A Neural Space-Time Representation for Text-to-Image Personalization. SIGGRAPH 2023.

---

> > ### Comment · Reviewer_pRpB · 2023-11-22
> > **Thanks for the detailed response**
> >
> > I appreciate the authors for your elaborate reply. Some of my concerns have been well addressed, but there are still some issues.
> >
> > - From Fig.13 of Appendix F.10, one observes that a customized cat is decomposed into several redundant words, i.e., cat, cats, kitty, kat, and so on. In my opinion, the rank-based selection strategy in Alg.1 should choose orthogonal words as far as possible. Could the authors explain this?
> > - For encoder-based methods mentioned above[1,2], [1] can be applied to open-domain scenarios and is suitable for the experiments of this paper. Therefore, the authors are also encouraged to compare with this method.
> > - Human face domain is very challenging for personalization of image diffusion models, which contains more perceptible details than other domains. Therefore, it is important to verify the effectiveness of proposed method in this domain.
> >
> >
> > [1] Yuxiang Wei, et al. Elite: Encoding visual concepts into textual embeddings for customized text-to-image generation. ICCV 2023.
> >
> > [2] Rinon Gal, et al. Encoder-based Domain Tuning for Fast Personalization of Text-to-Image Models. SIGGRAPH 2023.

---

> > > ### Author Response · Authors · 2023-11-22
> > > **Additional Response to Reviewer pRpB**
> > >
> > > Q1: From Fig.13 of Appendix F.10, one observes that a customized cat is decomposed into several redundant words, i.e., cat, cats, kitty, kat, and so on. In my opinion, the rank-based selection strategy in Alg.1 should choose orthogonal words as far as possible. Could the authors explain this?
> > >
> > > R1: We have shown ablation study to verify the choice of the vector distance in Appendix F.4. It can be seen that we have chosen dot product since it provides better results. Thus, it will choose basis vectors with similar direction and modal length in the textual space. Due to page limit, we have only shown 10 basis vectors in Figure 13 in Appendix F.10. However, the target vector is composed by all chosen basis vectors and can be seen as a reasonable combination of them. Thus, the redundance seems not to be a problem.
> > >
> > > Q2: For encoder-based methods mentioned above[1,2], [1] can be applied to open-domain scenarios and is suitable for the experiments of this paper. Therefore, the authors are also encouraged to compare with this method.
> > >
> > > R2: Thanks for correction. We have performed qualitative comparison with [1] using example ``cat''. Results have been added in Appendix F.9. It can be seen that although encoder-based method
> > > finetunes much faster (it only needs one forward step), it lacks text-to-image alignment ability in
> > > some cases (e.g., miss “sleeping” in the second case). Due to time limit, we have provided comparison with limited cases. We will add more comparisons with encoder-based methods in the next revision.
> > >
> > > Q3: Human face domain is very challenging for personalization of image diffusion models, which contains more perceptible details than other domains. Therefore, it is important to verify the effectiveness of proposed method in this domain.
> > >
> > > R3: Thanks for your opinion. We do realize the importance of human face personalization. However, human face personalization is rare in prior works. Thus, we take the lecun face in [2] as an example. Results can be seen in Appendix F.11. It can be seen that the proposed method generates results following the textual prompt while successfully reconstructing the input human face image. We will provide more qualitative results in the next revision.
> > >
> > > [1] Yuxiang Wei, et al. Elite: Encoding visual concepts into textual embeddings for customized text-to-image generation. ICCV 2023.
> > >
> > > [2] Rinon Gal, et al. Encoder-based Domain Tuning for Fast Personalization of Text-to-Image Models. SIGGRAPH 2023.

---

> > > > ### Comment · Reviewer_pRpB · 2023-11-22
> > > > **Thanks for the authors' feedback**
> > > >
> > > > Thank you for your additional experiments and clarification. My concerns have been well addressed, and I am glad to raise the score to 6.

---

### Official Review · Reviewer_s24C · 2023-11-03

**Soundness:** 3 good
**Presentation:** 3 good
**Contribution:** 2 fair
**Rating:** 3
**Confidence:** 4

**Summary:**

The paper introduces a method for learning embeddings in a low-dimensional textual subspace to achieve improved time efficiency and better preservation of text similarity in the learned embeddings for personalized text-to-image generation.

**Strengths:**

1. The writing is clear and easy to follow.

2. The authors propose using a reduced number of vectors to represent the original embedding, enabling the proposed model to undergo fewer training steps without significantly compromising performance.

3. The experiments presented in the paper support the authors' assertions. They require fewer training steps while achieving competitive performance compared to other methods.

**Weaknesses:**

1. I remain unconvinced regarding the claimed time efficiency of the proposed method. First, the paper only provides the number of training steps, but it lacks information on the actual time required for each step of the proposed method. Furthermore, the method's need for a loop to search for the proper number M, as shown in Algorithm 1, can be time-consuming. Consequently, the quantitative comparison in Table 2 does not demonstrate a significant advantage of the proposed method.

2. I also have reservations about the qualitative performance of the proposed method. The paper showcases only a limited number of results with restricted diversity, such as a limited variety of style images.

3. I am unclear as to why the authors assert that previous methods solely focus on image reconstruction, thereby degrading their ability to combine the learned embeddings with different textual prompts. The primary objective of personalized text-to-image generation is to create new images based on input images. Additionally, Textual Inversion can also combine various textual prompts. More comprehensive details and discussion are needed to support this claim.

**Questions:**

Please see above weaknesses. I am willing to change my rating if authors could address my concerns.

---

> ### Author Response · Authors · 2023-11-19
> **Response to Reviewer s24C**
>
> Q1: I remain unconvinced regarding the claimed time efficiency of the proposed method...
>
> R1: We have revised Table 1 to show the total training time compared with other three methods. It can be seen that our method is 6 times faster than the baseline method TI and achieves competitive results when compared with Dreambooth and Custom-Diffusion. Also, although it requires users to define the number $M$, we have experimentally observe that the embedding matrix $V_M$ in Algorithm 1 is highly linearly-independent, thus further releasing the constraint of checking the rank $r(V_M)$. Furthermore, it can be seen from Table 2 that BaTex outperforms TI in text-to-image alignment score with no performance degradation in reconstruction. Compared with DB and CD, it clearly shows that we reach the state of the art performance in text-to-image alignment score with competitive image-to-image alignment score. Thus, it can be concluded that BaTex achieves high alignment ability with significantly less computational overhead.
>
> Q2: I also have reservations about the qualitative performance of the proposed method...
>
> R2: Due to page limit, we have only provided limited results in the main text. However, diverse results have been placed in Appendix F, including ablation studies, qualitative results about more objects and styles and quantitative comparisons. Please head there for details.
>
> Q3: I am unclear as to why the authors assert that previous methods solely focus on image reconstruction...
>
> R3: As observe in the previous work [DreamBooth  Fine Tuning Text-to-Image Diffusion Models for Subject-Driven Generation] and [Multi-Concept Customization of Text-to-Image Diffusion], TI often get lower text-to-image alignment score due to its nature of only training the vector by reconstruction objective. As can be seen in Figure 2, when input the same complex text prompt to all four methods, BaTex generates reasonable results according to the input text conditions while TI occurs missing some objects mentioned, including ''lady'', "student'' and "table''. Thus, one of our motivations is to represent the word vector in a textual subspace to enhance its ability to combine with other words with less computational overhead.

---

### Official Review · Reviewer_qu14 · 2023-11-05

**Soundness:** 3 good
**Presentation:** 3 good
**Contribution:** 2 fair
**Rating:** 6
**Confidence:** 4

**Summary:**

This paper proposes a  method named as  BaTex for learning arbitrary embedding in a low-dimentional textual subspace. This paper also proposes an efficient selection strategy for determining the basis vectors of the textual subspace. The proposed methods achieve good performances on the public datasets.

**Strengths:**

This paper proposes a method named as BaTex for learning arbitrary embedding in a low-dimentional textual subspace, which is time-efficient and better preserves the text similarity of the learned embedding. The learned embeddings can not only faithfully reconstruct the input image, but also significantly improve its alignment with different textual prompt.

**Weaknesses:**

The novelty is limited. Although this paper proposes a method to extract the specific textual subspace for personalized text-to-image generation, the novelty of te proposed method is limited. The textual subspace vector is widely adopted for the conditional diffusion models, e.g. [Medical diffusion on a budget: textual inversion for medical image generation], [LaDI-VTON: Latent Diffusion Textual-Inversion Enhanced Virtual Try-On], etc. What's more, the pritority and advantages of the proposed method comparing with the traditional textual inversion is not obvious. The difference of the training step need to be clarified more clearly.

**Questions:**

1. Please highlight the priority and novelty of the proposed method from the whole diffusion procedures.
2. Please add more analysis of the training details of the proposed methods comparing with the textual inversion.
3. Please add more experimental results of more conditions of the text-to-image tasks.

---

> ### Author Response · Authors · 2023-11-19
> **Response to Reviewer qu14**
>
> Q1: The novelty is limited...
>
> R1: First of all, we claim that BaTex is the first method to introduce a specific textual subspace for personalization. Due to the slow training time of the baseline methods, which requires nearly an hour to train a single object, text-to-image personalization is ineffective to be applied in large-scale applications. In this paper, we propose a simple yet effective method to search the embedding in a subspace, which results in significant reduced training time and fast convergence speed. For the two paper [Medical diffusion on a budget] and [LaDI-VTON] mentioned above, they apply the personalization mechanism to other fields rather than improving the baseline methods. Thus, the novelty of the proposed method can be guaranteed by its novel way to represent and optimize the word vector in a textual subspace. Since our method offers fast convergence speed compared to baseline method, it naturally requires less training steps.
>
> Q2: Please highlight the priority and novelty of the proposed method from the whole diffusion procedures
>
> R2: As stated in R1, we are the first to propose a method to represent and optimize the word vector in a meaningful textual subspace. From the perspective of the whole diffusion procedure, we introduce a mechanism to optimize the word vector $v$ in textual subspace, which represents a pseudo-word. The vector $v$ is then concatenated to the embedding matrix $V_M$ and input into the text encoder to get the final condition $y$, and input into the noise predictor $\epsilon_{\phi}(z_t;\epsilon, t, y)$ to predict the corresponding noise in the reverse process at timestep $t$. Since the original personalization method needs up to an hour to optimize $v$ for each object, the proposed method significantly improves the training speed with no performance degradation.
>
> Q3: Please add more analysis of the training details of the proposed methods comparing with the textual inversion.
>
> R3: In Textual Inversion, it represents and optimizes the target vector in the original textual space, while in BaTex, a specific subspace is established and utilized using the proposed selection strategy. In each iteration, the trainable weights $w$ are updated using stochastic gradient descent with other modules frozen. After training, the weights are combined by the basis vectors to obtain the target word vector, which lies in the textual subspace.
>
> Q4: Please add more experimental results of more conditions of the text-to-image tasks.
>
> R4: Experimental results of more conditions have been added in Figure 4 and 5 in the supplementary material. As can be seen, the proposed method successfully reconstruct the input concept while generating diverse conditions following the provided text prompt.

---

### Meta-Review · Area_Chair_peYE · 2023-12-04

**Metareview:**

This paper extends the textual-inversion technique in text-to-image diffusion models. In particular, instead of optimizing the textual embedding in a free manner, the proposed method optimizes the textual embedding in a textual meaningful subspace. The authors claim this proposed method can enhance efficiency of textual inversion and has conducted experiments to demonstrate.

**Justification For Why Not Higher Score:**

- the effectiveness of the proposed method is not well justified. From the experiments, it is not clear whether the proposed method indeed improves either efficiency or quality of the generated images. As mentioned by the reviewer, "the pritority and advantages of the proposed method comparing with the traditional textual inversion is not obvious. The difference of the training step need to be clarified more clearly."

- the basis vector selection strategy is cumbersome and time-consuming.

**Justification For Why Not Lower Score:**

NA

---

### Decision · Program_Chairs · 2024-01-16

Reject